# Exchange pathways of plastoquinone and plastoquinol in the photosystem II complex

Floris J. Van Eerden[1], Manuel N. Melo[1,2], Pim W.J.M. Frederix[1], Xavier Periole[1] & Siewert J. Marrink[1]

Plastoquinone (PLQ) acts as an electron carrier between photosystem II (PSII) and the cytochrome $b_6f$ complex. To understand how PLQ enters and leaves PSII, here we show results of coarse grained molecular dynamics simulations of PSII embedded in the thylakoid membrane, covering a total simulation time of more than 0.5 ms. The long time scale allows the observation of many spontaneous entries of PLQ into PSII, and the unbinding of plastoquinol (PLQol) from the complex. In addition to the two known channels, we observe a third channel for PLQ/PLQol diffusion between the thylakoid membrane and the PLQ binding sites. Our simulations point to a promiscuous diffusion mechanism in which all three channels function as entry and exit channels. The exchange cavity serves as a PLQ reservoir. Our simulations provide a direct view on the exchange of electron carriers, a key step of the photosynthesis machinery.

[1] Groningen Biomolecular Sciences and Biotechnology Institute & Zernike Institute for Advanced Materials, University of Groningen, Nijenborgh 4, 9747 AG Groningen, The Netherlands. [2] Instituto de Tecnologia Química e Biológica António Xavier, Universidade Nova de Lisboa, Av. da República, 2780-157 Oeiras, Portugal. Correspondence and requests for materials should be addressed to S.J.M. (email: s.j.marrink@rug.nl).

Photosynthetic organisms convert light into chemical energy. This fundamental process involves four major protein complexes: photosystem II (PSII), cytochrome b₆f complex (Cyt b₆f), photosystem I and ATP synthase[1,2]. The process starts at PSII, which extracts electrons from water. The electrons travel subsequently to Cyt b₆f and PSI, after which they reduce $NADP^+$. The electrons are transported between these protein complexes by charge carriers. Plastoquinone (PLQ) is the charge carrier responsible for the electron transport from PSII to Cyt b₆f. Upon photoactivation of PSII, PLQ is double reduced and takes up two protons to become plastoquinol (PLQol) (Fig. 1a).

The PSII core complex, a homodimer consisting of 27 subunits in plants and 20 in cyanobacteria[3–5], coordinates two PLQs per monomer, named $Q_A$ and $Q_B$, symmetrically positioned around a non-heme iron (Fig. 1b,c). $Q_A$ is stationary and does not leave the protein; it just passes the electron on to $Q_B$. $Q_B$, however, leaves the protein after its conversion to PLQol; then a new $Q_B$ enters the binding site and the process can start again. In the X-ray structure of Guskov et al.[3] a third PLQ is present, coined $Q_C$. $Q_C$ is however not found in the later Umena and Wei structures[4,5]. The $Q_C$ site is located close to the $Q_B$ site, but the role of $Q_C$ is still highly debated.

While the $Q_A$ site is well buried in the core of the PSII complex, the $Q_B$ and $Q_C$ sites connect to a small cavity located within the protein (Fig. 1c). This so-called PLQ/PLQol exchange cavity is filled with lipids: on the stromal side with the negative charged phosphatidylglycerol (PG) and sulfoquinovosyldiacylglycerol (SQDG) lipids and on the lumenal side with digalactosyldiacyl-glycerol (DGDG) and monogalactosyldiacylglycerol (MGDG) lipids[4]. Two channels link the cavity to the thylakoid membrane[3]. Channel I, containing the tail of $Q_C$, is flanked by cyt b559α and PsbJ and opens up to the centre of the thylakoid membrane. The $Q_C$ head group protrudes into the PLQ exchange cavity, where it interacts with lipid and cofactor tails, but not with any amino acids. Channel II, containing the $Q_B$ tail, is located between the D2 subunit and cyt b559β, opening up more on the stromal side compared to channel I. The $Q_B$ headgroup is located close to the non-heme iron.

The presence of these two channels led to three different models for PLQ/PLQol diffusion involving the $Q_B$ and $Q_C$ sites[3]. In the 'alternating' mechanism, channels I and II are used both as an entry and as an exit. Each PLQol leaves however through the same channel as through which it entered as a PLQ. In the 'wriggling' mechanism, PLQ enters via channel I and PLQol leaves via channel II. This would be in line with the fact that PLQol is more polar than PLQ, preferring to leave through channel II which opens up closer to the membrane surface[6,7]. In the 'single channel' mechanism, only channel II is used and channel I is occupied by a stationary PLQ molecule ($Q_C$) that might be involved in redox reactions with cyt b559 (refs 8,9).

Here we present the results of coarse grained (CG) molecular dynamics (MD) simulations of the diffusion of PLQ and PLQol in and out of the cyanobacterial PSII complex. The use of a CG model enables simulations of multi-meric protein assemblies in a complex membrane environment, exploring time scales in the microsecond range[10–14]. Recently, we described the dynamics of the PSII complex including all of its cofactors, embedded in a realistic description of the thylakoid membrane based on the CG Martini model[15]. We extended this work, and now focus our analysis on the exchange pathways of PLQ and PLQol by quantifying the role of the different channels. Previous computational work has focused on the energetics of the $Q_A$ and $Q_B$ binding sites[16], and on simulation of PLQ either in solution[16] or in the thylakoid membrane[17]; for a review see ref. 18. An atomistic simulation has recently been performed on the difference in binding of PLQ/PLQol at the $Q_B$ site[19], but the simulation time was too short to observe any unbinding events. To our knowledge no simulations have been performed to address the full exchange of PLQ/PLQol in and out of PSII, which is the topic of the current manuscript.

We simulated the PSII dimer complex of the cyanobacteria *T. vulcanus*, including all cofactors and embedded in a realistic representation of the thylakoid membrane composed of MGDG, DGDG, SQDG and PG lipids (Fig. 2). To probe the exit and entry pathways of PLQ and PLQol, the $Q_B$ site was initially occupied by a PLQol, and PLQs were placed in the bulk membrane at about 5 mol%. Five replicate simulations were performed for a time period between 80 and 100 μs each (see Methods for details). We observe multiple binding and unbinding events of the

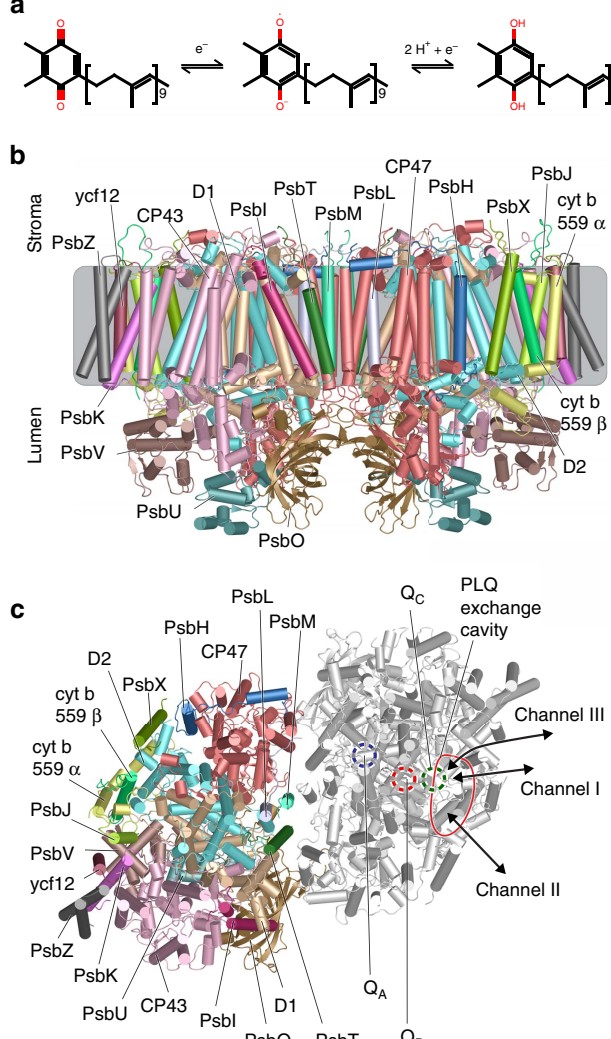

**Figure 1 | PSII and the PLQ exchange channels.** (a) Structure of plastoquinone, plastosemiquinone and plastoquinol. The uptake of one electron by plastoquinone results in the radical semiplastoquinone, an additional electron and two protons result in PLQol. (b) View on PSII dimer from the plane of the membrane, with labelling of the 19 subunits coloured according to chain. The grey box roughly indicates the position of the thylakoid membrane. (c) Stromal view on the PSII dimer. The left monomer is coloured and labelled as in b. In the right monomer the protein is coloured transparent white and the PLQ binding sites $Q_A$ (blue), $Q_B$ (red) and $Q_C$ (green) are indicated, as well as the two known channels I, II and the new channel III.

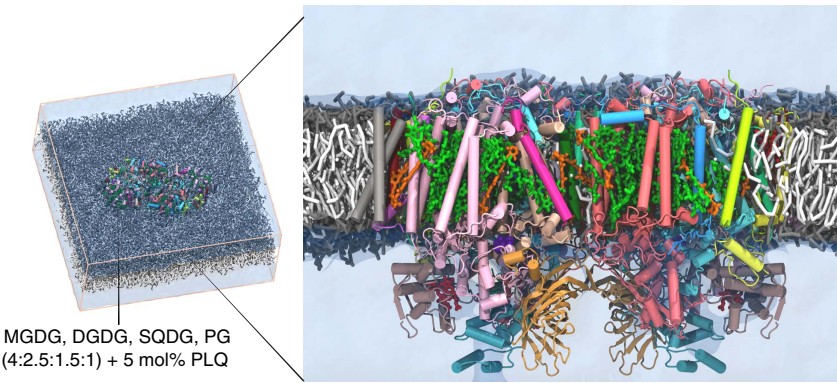

MGDG, DGDG, SQDG, PG
(4:2.5:1.5:1) + 5 mol% PLQ

**Figure 2 | Simulated system setup.** Snapshot of the simulation box with the inset showing the equilibrated PSII dimer with all cofactors embedded in the thylakoid membrane with composition as indicated. PSII is coloured as in Fig. 1b and the cofactors as follows: Chlorophyll a in green, pheophytin in blue, haem in red, PLQ in yellow, β-carotene in orange, oxygen evolving complex in purple. The headgroups of the thylakoid lipids are coloured grey and the tails white. Solvent is rendered as a transparent blue surface.

electron carriers, and discover a new pathway, denoted channel III (Fig. 1c), by which PLQ and PLQol can enter or leave the PSII complex.

## Results

**PLQol leaves the $Q_B$ site on a microsecond time scale**. In our previous simulations of the PSII dimer we showed that the oxidized form of the electron carrier, PLQ, remains stably bound at the $Q_B$ site[15]. We anticipate that the reduced form, PLQol, should leave the binding site in order to deliver two electrons to the cytochrome $b_6f$ complex. In line with our expectations, we observe the spontaneous unbinding of PLQol from the $Q_B$ site. Out of ten PLQols (two per dimer), five of the PLQols leave the $Q_B$ site completely during the simulation. Out of these five, one PLQol escapes the protein ending up in the bulk thylakoid membrane, two end up in the PLQ exchange cavity, and two remain trapped in the channels. The other five PLQols stay largely in place, with only transient or partial unbinding events being observed (for example, rapid rebinding, or unbinding of the head group but not of the tail). Individual unbinding times vary greatly, from within 0.1 to tens of microseconds, pointing out the stochastic nature of the unbinding process (see Supplementary Fig. 1). Note time scales in CG MD are only approximate; please see the SI for a more detailed discussion on their interpretation.

**PLQ enters exchange cavity and reaches binding sites**. To investigate how PLQs diffuse into the PLQ exchange cavity, we calculated a PLQ density map, combining the data from the five replicate simulations that contained additional PLQ molecules (138 molecules, ∼5 mol%) in the surrounding bulk thylakoid membrane. The resulting density map is visualized in Fig. 3a, showing the areas around the PSII dimer where it is more likely to encounter a PLQ during the simulation.

The first remarkable feature is the non-homogeneous distribution of PLQ around the protein. PLQ clearly has a preferred region of interacting with PSII, ranging approximately from subunits PsbZ to PsbH, with a few spots around CP43 and D1 where the PLQ tries to penetrate PSII. Remarkably, there is no density in and around the dimer cleft. It appears that PLQs accumulate at the side of the protein with access to the exchange cavity.

The exchange cavity itself is also clearly visible in the density map. This implies that PLQs spontaneously enter the exchange cavity from the bulk thylakoid membrane. Interestingly the density is not homogenously distributed within the cavity, but located more towards the cyt b559 and away from CP43 (Fig. 3a).

At the end of the simulation the amount of PLQ inside the exchange cavity is $1.0 \pm .2$ (s.e.m., $n = 10$) PLQ molecules. It is likely that this is still an underestimation, as equilibration of the PLQ population in the exchange cavity is a slow process. Note that, in most simulations, a PLQol molecule is also present, either at the $Q_B$ site or trapped in the exchange cavity.

Despite the presence of PLQ in the exchange cavity we did not observe in any of the simulations PLQ docking into the $Q_B$ site, which would in principle be possible in the cases in which PLQol has left. This is reflected by the lack of density right at the $Q_B$ site. The $Q_C$ site, on the other hand, is frequently visited by PLQs as is apparent from the density map. In addition, the $Q_A$ site, containing the stationary PLQ molecule, is clearly visible. Visual inspection of the trajectories shows that subtle side chain reorientations, especially of D1-HIS215, that occur after unbinding of PLQol, prevent easy access to the $Q_B$ binding site. In order to increase the chance of a PLQ binding to the $Q_B$ site, more than 50 μs of adaptive MD was performed (see Methods). During the adaptive MD simulations PLQ could closely approach the binding position found in the crystal structure, although with a slightly different orientation of the PLQ headgroup (see Supplementary Fig. 2).

**Discovery of a third exchange channel**. Interestingly, the PLQ density map shows the existence of three clear distinct pathways connecting the membrane to the PLQ exchange cavity, indicated by arrows in Fig. 3a. These pathways correspond to channels I and II previously reported in the literature[3], as well as a novel pathway, denoted channel III. Our data thus imply the existence of a third exchange channel. The novel channel emerges when PsbJ moves towards cyt b559α while PsbK and ycf12 move in the opposite direction, creating a tunnel between PsbJ on one side and PsbK and ycf12 on the other side.

To get an estimate about the PLQ fluxes through the three channels we counted how many PLQs pass through each channel in the five different simulations (see Supplementary Methods, Supplementary Table 1). The data are shown in Table 1, averaged over the ten monomers (results for the individual monomers are given in Supplementary Table 2). In total, we observed 19 full entries and 11 full exits, over the 0.5 ms aggregate simulation time. In addition, many additional PLQs are found trapped inside the channels at the end of the simulation.

Our data furthermore show that each of the channels is used both as an entrance and as an exit to the PLQ cavity. Sometimes a different channel is used to leave the cavity than the channel used

to enter. It is however also common that the same channel is used. In most of the complexes, the entry flux is higher than the exit flux (Table 1), causing the net increase in the amount of PLQs present in the PLQ exchange cavity (see above). On average, every 33 µs a cofactor enters or leaves one of the two PLQ cavities.

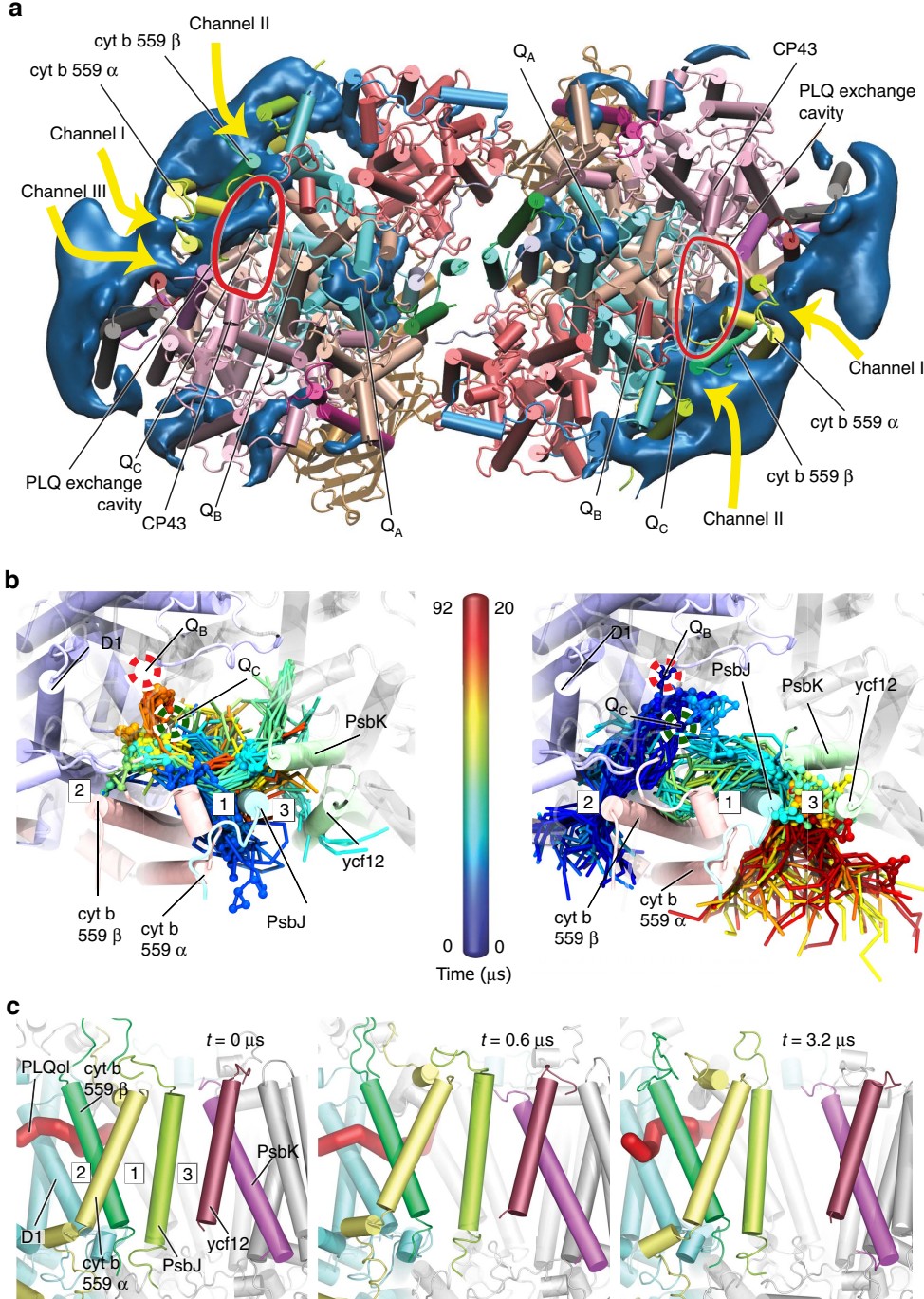

**Figure 3 | Diffusion of PLQ and PLQol in and out of PSII.** (**a**) Stromal view on PSII with occupancy density of PLQ from the five replicate simulations. Indicated are the locations of the headgroups of $Q_A$, $Q_B$ and $Q_C$, as well as channel I, channel II and channel III. The red line approximately indicates the location of the PLQ exchange cavity. Note that in the right monomer channel III is not visible at the chosen threshold level. (**b**) Time series of snapshots of two diffusion events of a single PLQ into PSII (left) and a single PLQol out of PSII (right); the PLQ and PLQol headgroup beads are shown as spheres for clarity. The red dashed circle indicates the location of the $Q_B$ binding site and the green dashed circle the location of the $Q_C$ headgroup. In the left panel the PLQ enters the complex via channel I. After entry the PLQ diffuses around in the PLQ exchange cavity, its tail briefly sticking out of channel III. By the end of the simulation the PLQ headgroup is very close to the $Q_B$ binding site. In the right panel the PLQol moves from its initial position at the $Q_B$ site with its tail inside channel II, towards the PLQ exchange cavity. Subsequently its headgroup moves into channel III, with the PLQ tail still in the exchange cavity. The headgroup remains in the channel opening, while the PLQol tail folds over the headgroup into the membrane through channel III. Eventually the molecule diffuses out of the channel into the membrane. (**c**) Snapshots of PSII from the side showing the increase of the channel III opening due to the movement of the PsbJ and ycf12 subunits.

**Table 1 | Comparison of the flux\* of PLQ through different channels.**

|  | Channel I | Channel II | Channel III |
|---|---|---|---|
| Flux in | 10 ± 2 | 1 ± 1 | 8 ± 2 |
| Flux out | 4 ± 3 | 1 ± 1 | 6 ± 2 |

*The average flux of PLQs (no. of molecules ms$^{-1}$ per monomer), diffusing in and out of the two PLQ exchange cavities through the three channels, is shown. Standard errors are given based on ten independent measurements (five simulations, two monomers). More details are shown in Supplementary Table 2.

Comparing the fluxes between the channels, it appears that channels I and III are used by PLQ with roughly the same frequency, but significantly fewer cofactors pass through channel II. In the latter channel, they often get stuck and do not completely pass through the channel (see also Supplementary Table 2), which we attribute to the smaller size of channel II. Channel I has dimensions of $1.0 \times 2.0 \, \mathrm{nm}^2$, while the dimensions of channel II are only $1.0 \times 1.2 \, \mathrm{nm}^2$ (ref. 20). We estimate the size of channel III as $1.1 \times 2.0 \, \mathrm{nm}^2$, similar to channel I (see Supplementary Methods). During the simulation, the shape and size of the channels fluctuates significantly, with sizes transiently increasing or decreasing up to 100%.

To explore whether PLQols can also enter the cavity using the same channels, six additional simulations were performed (aggregate time 0.25 ms) with PLQol at 5 mol% in the bulk membrane (see Supplementary Methods). The results (Supplementary Tables 3 and 4) indicate a very similar behaviour as observed for PLQ, implying that PLQol is potentially able to (re)enter the PLQ exchange cavity.

Visual inspection of our simulation trajectories reveals that the subunits lining the channels are dynamic, and can move with respect to each other. In particular, there is movement of the helices cyt b559α + β, PsbJ, PsbK ycf12 and PsbX. These relative movements can either result in constriction or even complete closure of a channel, or result in an increased capacity of a channel due to a larger channel opening. This is especially noticeable for channels I and III. Graphical snapshots illustrating the entrance of PLQ through channel I, the exit of PLQol through channel III and the opening of channel III are shown in Fig. 3b,c. See also Supplementary Movies 1–3.

**PLQs can reorient inside the exchange cavity.** Also of interest is the observation that both PLQs and PLQols can enter the channels in two orientations, headgroup first or tail first (see Supplementary Fig. 3). Inside the exchange cavity, PLQ can further reorient by flip-flopping between the stromal and lumenal leaflets. We observed 14 such events during the 0.5 ms aggregate time of the five replicate simulations. Taking into account the amount and time PLQ molecules spent inside the cavity, this corresponds to a flip-flop time scale of the order of 100 μs. Compared to the typical flip-flop time of PLQ in the thylakoid bulk membrane, which is around 1 μs as estimated from simulations using the same CG model[17], reorientation dynamics inside the cavity is thus slowed down by two orders of magnitude. In case of PLQol, no flip-flop events were observed for the PLQol molecules inside the cavity, in any of the six replicate simulations. Again taking into account the amount and time spent in the cavity, we estimate a lower bound of the reorientation time scale of 200 μs. The lower flip-flop rate of PLQol compared to PLQ is in line with the lower flip-flop rate for PLQol versus PLQ in bulk membranes (at least one order difference[17,21]), and can be attributed to the more polar nature of the reduced cofactor.

## Discussion

We have investigated the behaviour of PLQ and PLQol in PSII, based on coarse-grained molecular dynamics simulations. The CG approach allowed us to simulate the full PSII dimer system, including all cofactors and embedded in a realistic thylakoid membrane environment, on an aggregate time scale of almost 0.5 ms. We find that PLQs accumulate around the PSII complex at sides close to the exchange cavity, and are able to enter and leave this cavity using three different channels. Of these, two channels correspond to the known channels I and II. The third channel is a novel channel that has not been reported before. Taken together, our simulations point to a plastic behaviour of the PLQ exchange channels. Here we discuss our results in light of the current literature view on PLQ binding and exchange pathways. A discussion of the limitations of our approach can be found in the Supporting Information (Supplementary Note 1). In light of these limitations, it is imperative that our results are eventually verified by more detailed all-atom models, and/or validated experimentally.

In our simulations with PLQ occupying the $Q_B$ site, both $Q_A$ and $Q_B$ remain stationary. This is in line with their function in photosynthesis. Experiments suggests that it is relatively difficult to remove $Q_A$ from its binding site[22–24]. The $Q_B$ PLQ is only expected to leave the site after being converted to PLQol. This is confirmed by our simulations: we observe that while all PLQs remain in the $Q_B$ pocket, PLQols diffuse out of this binding site. The affinity of PLQ and PLQols for the $Q_C$ site appears weak. Although the site is frequently visited by PLQ/PLQols diffusing through channel I, actual binding is not observed. We hypothesize that the $Q_C$ site in the crystal structure of Guskov et al.[3] either originates from a PLQ trapped inside the channel under the crystallization conditions, or represents a number of weaker binding spots around the $Q_C$ site.

The simulations reveal that there are preferential regions on PSII where PLQ interacts, and from which PLQ can enter the exchange cavity. Importantly, our data suggest that the PLQ cavity could function as a PLQ reservoir. We quantified a gain of about one PLQ molecule per monomer compared with the Umena structure in which only $Q_B$ is present[4]. Considering our simulations did not reach equilibrium yet, the equilibrium population could be even higher. The PLQ exchange cavity can thus function as a kind of local reservoir of PLQs, where PLQs can reorient. The latter is especially relevant because some of the PLQs enter tail first. Note that the PLQ cavity reservoir should not be confused with the total PLQ pool, which is likely located for a large part outside the PSII complex in the thylakoid membrane, and estimated to contain between 9 and 30 PLQs per complex[25,26]. The entry and exit kinetics of PLQ, on average one PLQ per 33 μs per monomer, is fast compared to both the first reduction step of PLQ at the $Q_B$ site towards semiplastoquinone, that has a time constant of a few hundreds of microseconds, and the second redox step that shows somewhat slower kinetics[26,27]. This implies that a PLQ is always available close to the $Q_B$ site to replace PLQ upon its reduction to PLQol, at least when the PLQ pool is largely oxidized. Interestingly, the simulations with excess PLQol in the thylakoid membrane showed that the reduced electron carrier is able to potentially re-enter the exchange cavity, competing with PLQ. This could be an important regulation mechanism to reduce reduction rates under conditions when PSII is operating too fast and the PLQ pool becomes reduced.

Our simulations provide direct evidence for the existence of designated PLQ/PLQol exchange pathways. Surprisingly, we find exchange taking place via three distinct channels, rather than two channels that are thus far assumed. Apart from channel I and channel II that have been described before[3], we observed a third

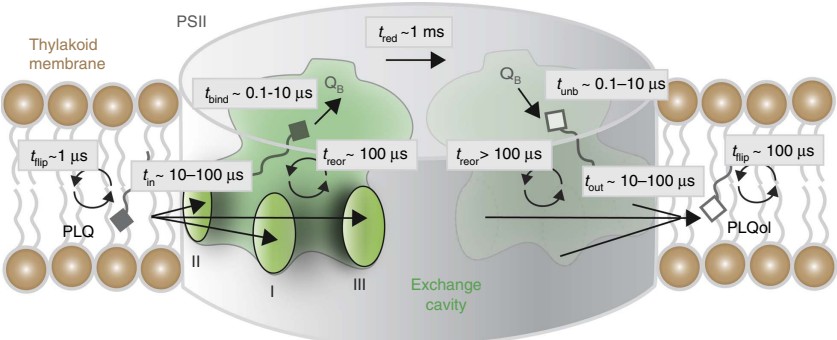

**Figure 4 | Promiscuous exchange model.** A PLQ can enter through any of the three channels to the PLQ exchange cavity. In the cavity the PLQ can reorient and bind to the $Q_B$ site to get reduced towards PLQol. Subsequently PLQol can diffuse via the cavity into the thylakoid membrane through any of the three channels. Reorientation is possible in the membrane environment as well. The indicated time scales of the individual steps are estimates based on the current work (channel entrance $t_{in}$ and exit $t_{out}$ times, reorientation time $t_{reor}$ inside the exchange cavity, and binding/unbinding time scales $t_{bind}$, $t_{unb}$ to $Q_B$ site), on previous simulations[17,21] (flip-flop time $t_{flip}$ inside the bulk thylakoid membrane), or on existing experimental data[26,27] (time scale $t_{red}$ of PLQ reduction to PLQol). Note that the reduction step in reality takes place within the same $Q_B$ site.

channel between subunits PsbJ and ycf12/PsbK. The simulations show that channels I and III are used more or less equally, but that the narrower channel II is used significantly less. The subunits lining the three channels can undergo conformational changes, modulating the relative opening or closure of channels I and III in particular. Is channel III really used *in vivo*? Mutants studies have already shown that PsbJ is likely to be involved in the electron transport from $Q_A$ to the PLQ pool, pointing to the importance of cyt b559 for photosynthesis, and revealed that PsbX has an influence on PLQ turnover[28–34]. However, a way to really assess the existence of channel III *in vitro* could be by measuring the distance between the various helices over time. This could be achieved by the inclusion of fluorescent or electromagnetic probes in helices cyt b559α, PsbJ and ycf12 and subsequently reading out the distances between the subunits over time using FRET or EPR measurements. Closing channels, by crosslinking the constituent helices, might be another approach to study the existence and usage of the various channels. One might be able to close, for example, channel III by crosslinking PsbJ, ycf12 and PsbK together and subsequently measure the effect on the redox potential of $Q_A$ and the PLQ pool.

In the literature, three different PLQ exchange mechanisms have been proposed: the alternating, the wriggling and the single channel mechanism[3]. The simulations do not match any of them fully. Our data agree with the alternating mechanism in the sense that both channels are used as an entry and an exit. It differs to the point that the cofactors do not have to leave through the same channel as through which they entered. Our simulations agree with the wriggling mechanism to the point that channel I can be used for PLQ entry and channel II for PLQol exit. In the wriggling model, however, channel I is exclusively used as an entry and channel II exclusively as an exit. We observe that PLQs enter the PLQ cavity using all channels and that PLQol can leave the binding site also making use of all three channels. Our simulations do not agree with the single channel mechanism in which only channel II is actively used and channel I is occupied by a stationary PLQ. In our simulations, the flux through channel I is actually significantly larger. Our findings are therefore also in disagreement with the conclusions derived from the MD study of Zobnina *et al.*[19], apparently supporting the existence of a single channel model. Our simulations are however four orders of magnitude longer and Zobnina *et al.*[19] do not observe any actual exchange of PLQs or PLQols. Finally, in all three mechanisms the PLQs directly diffuse to the $Q_B$ site after entering a channel and directly leave the complex after being reduced to PLQol. In contrast, we observe the exchange cavity to serve as a PLQ

reservoir allowing for substantial PLQ rearrangements before/after binding/unbinding.

The model emerging from our simulations is one in which three channels exist, each serving as both entry and exit pathway for both PLQ and PLQol. The exchange cavity serves as a temporary PLQ reservoir in which the PLQs can reorient. We denote this model the 'promiscuous' mechanism, to be considered as alternative for the alternating, the wriggling and the single channel mechanisms. A schematic of the new model is shown in Fig. 4.

Taken together, based on large-scale simulations, we have been able to shed important light on the mechanism by which PLQs and PLQols can diffuse in and out of the PSII complex. Our simulations do not fully agree with any of the three diffusion mechanisms described in the literature[3]. Instead they point to a less organized, less deterministic model. Nine main observations can be made. (1) Three different channels exist that all can be used as an entry and an exit channel. (2) The entry and exit channel do not have to be the same for an individual PLQ; a number of PLQs enter and leave through the same channel, but others do this by a different channel. (3) The $Q_C$ site likely represents a weaker binding spot. (4) PLQs can pass through the channels in at least two different orientations, with their headgroup first or with their tail first. (5) PLQs do not directly dock at the $Q_B$ site from the channel, instead they first enter in the PLQ exchange cavity where they can diffuse around and reorient themselves. (6) PLQs can accumulate in the PLQ exchange cavity forming a PLQ reservoir. (7) The flux through channels I and III is more or less equal and several times larger than through channel II. (8) The relative flux through channels I and III is influenced by the relative conformations of cyt b559, PsbJ, ycf12 and PsbK. Possibly PsbX might be able to influence the flux through channel II. (9) The side of the protein where the channel openings are located acts as a funnel, accumulating PLQs towards the entrances of the exchange cavity.

The combination of these nine observations leads to a new model, the 'promiscuous' mechanism, in which channels I and III are primarily used, and each channel functions as an uncorrelated entry and exit of PLQ/PLQol. The PLQ exchange cavity can function as a local PLQ supply in which the PLQs and PLQols can reorient and there is a regulatory function of subunits cyt b559, PsbJ, ycf12 and PsbK.

## Methods
**System setup.** To study the dynamics of PLQ and PLQol exchange in the PSII complex, we used the equilibrated structure from our previous simulations of the PSII complex[15]. Here, we describe five independent simulations of the dimer in

which the $Q_B$ pocket contained a PLQol and extra PLQs were added to the thylakoid membrane.

The simulations are based on the crystal structure of the cyanobacterial PSII complex from Umena et al.[4] with PDB ID: 3ARC. The protein was coarse grained together with all of its cofactors and embedded in a thylakoid membrane composed out of 2,686 lipids using the insane script[35]. The membrane is composed of the negative charged PG and SQDG, and the neutral MGDG and DGDG lipids with oleoyl and palmitoyl tails, a realistic representation of the thylakoid membrane[17,36]. The system further contains 73,144 CG water beads (representing four times as many water molecules), 455 $Na^+$ and 455 $Cl^-$ ions (around 100 mM NaCl), plus 1032 $Na^+$ counter ions to neutralize the overall charge. More details of the setup of this system can be found in our recent publication[15].

The five different simulations described in the current work all started from the same initial structure, but with different seeds for the initial randomized velocities. In order to investigate the pathways of PLQ diffusion to the $Q_B$ binding site, 69 PLQ molecules were inserted into each membrane leaflet at the start of the simulation, totalling 138 free PLQs, yielding a concentration of about 5 mol % in the membrane. The PLQs were added to a pre-equilibrated bilayer by increasing the lateral dimensions of the box by 1.27 nm and adding 69 PLQs to each leaflet using the insane script[35]. The PLQs present in the $Q_B$ sites were modified to PLQols. The $Q_C$ site was left unoccupied.

**Simulation details.** The Martini force field version 2.2 (refs 37,38) in conjunction with the ElNeDyn elastic network[39] were used to model the interactions. The PLQ and PLQol parameters originate from ref. 21. The lipid parameters were taken from ref. 40 with the modification as described in ref. 17. GROMACS version 4.5.5 (ref. 41) was used to integrate the equations of motion with the common Martini settings for the Martini force field[42]. The simulations were run in the isothermal-isobaric (NpT) ensemble. Since T. vulcanus is a thermophile, the simulations were performed at 328 K, maintaining the thylakoid membrane in the fluid phase. The temperature was controlled using the V-rescale thermostat with a coupling constant of $\tau_t = 2.0$ ps (ref. 43). The pressure was semi-isotropically coupled to an external bath of $p = 1$ bar with a coupling constant of $\tau_p = 1.0$ ps and a compressibility of $\chi = 3.0^{-4} bar^{-1}$ using the Berendsen barostat[44]. A shifted potential with a cutoff of 1.2 nm in conjunction with a dielectric constant of 15 was used to model the electrostatic interactions. The Van der Waals interactions were also calculated using a shifted potential, with a cut off of 1.2 nm and a switch at 0.9 nm.

The five replicate simulations had a length of 84.2, 92.1, 97.4, 94.3 and 106.5 μs, summing up to almost 0.5 ms. Trajectories were saved and analysed every 500 ps, the first 1 μs being discarded as equilibration time. In order to increase the chance of a PLQ docking to the $Q_B$ site, an adaptive MD simulation approach was used consisting of an iterative process in which new, relative short simulations are spawn from promising configurations. Such an adaptive approach has been shown to be efficient in observing rare events, such as the binding of a ligand to its binding pocket[45]. Here, configurations were selected in which a PLQ was in a favourable condition to approach the bicarbonate ion. All together, we performed 41 independent short simulations totalling more than 50 μs simulation time. Details of the analysis, and additional simulations with excess PLQol, can be found in the Supporting Information (Supplementary Methods)[46,47]. Visual molecular dynamics (VMD) (ref. 46) and pymol (ref. 47) were used to generate the images.

**Data availability.** The authors declare that all data supporting the findings of this study are available within the manuscript and its supplementary files or are available from the corresponding author on request.

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

## Acknowledgements
We thank Albert Guskov for helpful discussions and critical reading of the manuscript, and Marina Guskova for help with the illustrations. F.J.V.E. acknowledges funding from the Foundation of Fundamental Research of Matter (FOM), and S.J.M. acknowledges funding through an ERC Advanced Grant 'COMP-MICR-CROW-MEM'.

## Author contributions
F.J.V.E., X.P. and S.J.M. designed the research; F.J.V.E. and M.N.M. performed the research; all authors analysed the data and wrote the paper.

## Additional information

**Competing interests:** The authors declare no competing financial interests.

**Publisher's note**: 

