## [Peer Review File · Nature Communications]

REVIEWERS' COMMENTS:

Reviewer #1 (Remarks to the Author):

The manuscript presents besides the two known exchange pathways a novel third exchange pathway of plastoquinone (PLQ) and plastoquinol (PLQol) in the photosystem II core complex (PSIIcc), which has been unraveled by coarse grained molecular dynamics simulations of PSIIcc-dimers embedded in the thylakoid membrane, covering a total simulation time of more than 0.5 milliseconds. The long time scale allows the observation of many spontaneous binding of PLQ, and unbinding of plastoquinol (PLQol). The presented simulations point to a promiscuous diffusion mechanism in which all three channels (PLQ/PLQol) serve as entry and exit channels. The third PLQ/PLQol channel of PSII is the most exciting part of the work.

The work is novel - no similar MD data (time scale) have appeared in the past. The work is sound and the authors are well qualified for such a study as is evidenced by their earlier work. The present data are important and are already overdue to elucidate the mechanism of the PLQ/PLQol exchange in PSII. This work represents a crucial starting point for experiments aiming at a verification of the discussed ideas.

The manuscript is well-written with clear figures and provides a wealth of supplementary information.

I strongly recommend publication, subject to a small correction:

On p. 3, lines 120/121, the authors refer to PLQ and PLQol as "reduced" and "oxidized", respectively. It should be the other way around.

Some of the references are incomplete.

Reviewer #2 (Remarks to the Author):

General

This paper describes coarse grained MD simulations of the Photosystem II aimed at understanding of the mechanism by which mobile electron carriers diffuse in and out of this complex. Plastoquinone diffusion is relatively slow process, and all-atom MD simulations of such scale are currently not possible. The manuscript describes long coarse grained simulations reaching 0.1 ms time which allowed the authors to observe multiple events of the reduced PQ unbinding and leaving the exchange cavity, as well as the oxidized PQ entering PSII and approaching the binding site. PLQ molecules accumulated in the exchange cavity entering and leaving it via all 3 previously found channels. While coarse grained simulations have many limitations, implications of these limitations are described in detail. The coarse grained force field has been thoroughly tested and seems to be appropriate for simulation of the system. The results of this study are important for understanding

mechanism of electron transfer and its regulation. Publication is recommended subject to minor amendments.

Specific

1. The static figures are excellent, but this is inherently a difficult 3D visualization that is being communicated. The work would be considerably more impactful if there were supplemental video files better conveying the dynamics to help readers visualize the results.
 2. The observation of PLQ accumulation near the exchange cavity is very interesting. It would be nice if the authors suggested what molecular features may be responsible for this non-homogeneous distribution.
 3. l.119: ".. reduced form of the electron carrier, PLQ, remains stably bound .." - PLQ is oxidized form, not reduced.
 4. l.120: "We anticipate that the oxidized form, PLQol, should leave the binding site" - PLQol is reduced form, not oxidized.
 5. l.256 "that is relatively difficult" should be "that it is relatively difficult"
 6. l.278-283. "The entry and exit kinetics of PLQ, on average one PLQ per 33 μ s per monomer, is fast compared to both the first reduction step of QB towards QB \cdot^- that has a time constant of a few hundreds of microseconds (26) and the minimum time required for reoxidation of QA. This implies that a PLQ is always available close to the QB site to replace PLQ upon its reduction to PLQol."
- QA is the oxidized form which is reduced to QA \cdot^- by electron transfer from pheophytin. The authors probably meant QA \cdot^- reoxidation, but it is the same reaction with the same kinetics as QB reduction. This should be clarified.
- The conclusion that "a PLQ is always available close to the QB site to replace PLQ upon its reduction to PLQol" is only valid in conditions when PLQ pool is fully oxidized. In reality under many physiological conditions PLQ pool becomes reduced to a certain degree. This should be clarified. It would be also useful to estimate at what percentage of reduced PLQ diffusion process becomes limiting for electron transfer.
7. l.727 typo, "..2 PLQols 140 PLQols.." - should be 140 PLQs
 8. Figure S2. It would be useful to show also the pose of the CG PLQ bound to the QB site.
- Reviewer #3 (Remarks to the Author):**
- The manuscript "Exchange Pathways of Plastoquinone and Plastoquinol in the Photosystem

II Complex" by F. J. Van Eerden et al. discusses the mechanism of how plastoquinone enters and leaves photosystem II, studied by coarse grained molecular dynamics simulations based on the Martini force field using GROMACS at 328K for a duration of several hundred microseconds. The manuscript predicts a promiscuous diffusion mechanism involving three channels that serve as entry and exit. Specific subunits are identified to play an important role in directing plastoquinone flux.

The manuscript is timely; describes a ubiquitous system of fundamental importance (photosystem 2) and a critical rate limiting energy conversion step therein (plastoquinone diffusion); employs a simulation method suitable to the timescale and complexity of the process. All in all, the manuscript is publishable. Some issues are listed below for the authors to consider.

In my opinion, the primary outcomes of the simulation could be exploited further in a quantitative description of the rate-limiting features of the diffusion channels and associated promiscuity property reported here. Instead, it appears that the flux computation is presented as a final result and its implications are not developed further. The flux computation is surely important, but its consequences for the primary function of photosystem 2 can likely be quantified, possibly alongside a coupling to the PLQ state of the neighboring thylakoid membrane.

Is it possible to construct a Markov State model out of the fluxes computed by this simulation? What implications can be derived regarding the connectivity topology of the channels and the rate limiting features of the diffusion processes observed?

I would like to note here, for the editor, that I do not consider the construction of a Markov State model or a rate kinetics analysis as a precondition for the acceptance of the manuscript. The authors may decide such an analysis to be beyond the current scope or better suited for a separate publication. However, it appears to me that the current manuscript could be strengthened by such an approach.

A potential concern is the ability of coarse-grained simulations to capture chemical details of the diffusion processes. Fortunately, the authors explicitly discuss the shortcomings of their approach, a commendable choice, stating clearly that the timescales reported can be regarded only as an order of magnitude estimates. Though it may be argued that an all-atom MD simulation would have had advantages over the coarse grained approach employed in the current study, the ~100 microsecond timescale of the diffusion process, along with the necessity of multiple simulations, somewhat justifies the current approach.

Minor point: Layout of Fig. 1. Simulation details in Fig. 1E are buried due to small size. I would suggest a larger width to display the details more prominently.

Fig. 3 could perhaps be improved by incorporating the flux results from Table 1, either as a

separate inset containing a state diagram or an overlay.

Replies to the specific referee comments (original comments in italics)

Referee 1:

I strongly recommend publication, subject to a small correction:

On p. 3, lines 120/121, the authors refer to PLQ and PLQol as “reduced” and “oxidized”, respectively. It should be the other way around.

Thanks, we fixed this issue.

Some of the references are incomplete.

References are now completed.

Referee 2:

1. The static figures are excellent, but this is inherently a difficult 3D visualization that is being communicated. The work would be considerably more impactful if there were supplemental video files better conveying the dynamics to help readers visualize the results.

Agreed, we now upload three movies to better visualize the dynamics.

2. The observation of PLQ accumulation near the exchange cavity is very interesting. It would be nice if the authors suggested what molecular features may be responsible for this non-homogeneous distribution.

We looked into this, but could not find an obvious driving force. Free energy calculations could be performed to assess this in more detail, but this is outside the scope of the current work.

3. l.119: *".. reduced form of the electron carrier, PLQ, remains stably bound .." - PLQ is oxidized form, not reduced.*

Thanks, fixed.

4. l.120: *"We anticipate that the oxidized form, PLQol, should leave the binding site" - PLQol is reduced form, not oxidized.*

Fixed.

5. l.256 *"that is relatively difficult" should be "that it is relatively difficult"*

Fixed

6. l.278-283. *"The entry and exit kinetics of PLQ, on average one PLQ per 33 μ s per monomer, is fast compared to both the first reduction step of QB towards QB⁻ that has a time constant of a few hundreds of microseconds (26) and the minimum time required for reoxidation of QA. This implies that a PLQ is always available close to the QB site to replace PLQ upon its reduction to PLQol." QA is the oxidized form which is reduced to QA⁻ by electron transfer from pheophytin. The authors probably meant QA⁻ reoxidation, but it is the same reaction with the same kinetics as QB reduction. This should be clarified.*

Agreed, and now better explained in the revised manuscript.

The conclusion that "a PLQ is always available close to the QB site to replace PLQ upon its reduction to PLQol" is only valid in conditions when PLQ pool is fully oxidized. In reality under many physiological conditions PLQ pool becomes reduced to a certain degree. This should be clarified. It would be also useful to estimate at what percentage of reduced PLQ diffusion process becomes limiting for electron transfer.

The possibility that the reduced form (PLQol) could actually compete with PLQ is interesting. We added results of additional simulations with excess PLQol that indeed show the entering of PLQols into the exchange cavity under those conditions. To provide a more quantitative estimate is difficult, and requires more simulations with different concentrations and ratios of the oxidized and reduced forms of the electron carriers.

7. l.727 typo, *"..2 PLQols 140 PLQols.." - should be 140 PLQs*

Thanks, fixed.

8. *Figure S2. It would be useful to show also the pose of the CG PLQ bound to the QB site*

This is already shown (in yellow). Legend has been adapted to make this more clear.

Referee 3:

In my opinion, the primary outcomes of the simulation could be exploited further in a quantitative description of the rate-limiting features of the diffusion channels and associated promiscuity property reported here. Instead, it appears that the flux computation is presented as a final result and its implications are not developed further. The flux computation is surely important, but its consequences for the primary function of photosystem 2 can likely be quantified, possibly alongside a coupling to the PLQ state of the neighboring thylakoid

membrane. Is it possible to construct a Markov State model out of the fluxes computed by this simulation? What implications can be derived regarding the connectivity topology of the channels and the rate limiting features of the diffusion processes observed? I would like to note here, for the editor, that I do not consider the construction of a Markov State model or a rate kinetics analysis as a precondition for the acceptance of the manuscript. The authors may decide such an analysis to be beyond the current scope or better suited for a separate publication. However, it appears to me that the current manuscript could be strengthened by such an approach.

We agree that such an analysis would be very interesting, but consider it as outside the scope of the current paper.

A potential concern is the ability of coarse-grained simulations to capture chemical details of the diffusion processes. Fortunately, the authors explicitly discuss the shortcomings of their approach, a commendable choice, stating clearly that the timescales reported can be regarded only as an order of magnitude estimates. Though it may be argued that an all-atom MD simulation would have had advantages over the coarse grained approach employed in the current study, the ~100 microsecond timescale of the diffusion process, along with the necessity of multiple simulations, somewhat justifies the current approach.

We agree that being aware of the shortcomings of our model is important. We extended the discussion of the limitations of our model, see the Supplementary Note 1.

Minor point: Layout of Fig. 1. Simulation details in Fig. 1E are buried due to small size. I would suggest a larger width to display the details more prominently.

Agreed, the figure has been improved.

Fig. 3 could perhaps be improved by incorporating the flux results from Table 1, either as a separate inset containing a state diagram or an overlay.

We incorporated the flux results into the revised figure (now Figure 4).